# STARFORMER: STRUCTURAL ATTENTION TRANSFORMER FOR LONG-TERM TIME SERIES FORECASTING

## ABSTRACT

In recent years, Transformers have been gaining attention in the fields of Natural Language Processing, Computer Vision and Time Series. Despite the lack of a mechanism to exploit the characteristics of time series data, it has demonstrated its potential in a variety of applications. These capability gaps, including lack of decomposability and interpretability, often make them suboptimal in long-term forecasting efforts. To address these issues, many recent studies show performance improvements by replacing self-attention with traditional time series decomposition algorithms or Fourier transform algorithms. This paper follows recent research trends. This paper introduces **ST**ructural **A**ttention t**R**ansformer, called **STARformer**, an innovative transformer architecture optimized for time series forecasting. In this work, we improve the transformer by replacing self-attention. This architecture obtains structural attention from a single-linear-layer model and amplifies efficiency and accuracy by replacing the self-attention of existing transformers. Our model i) proposes a methodology for easily solving complex time series, and ii) demonstrates excellent performance using structural attention based on future trends(or seasonal parts). STARformer, which replaces the existing transformer's self-attention with a structured attention block, outperforms the existing baselines by a non-trivial margin in experiments using 9 real datasets and 12 baselines.

## 1 INTRODUCTION

Time-series forecasting is important in various fields like managing energy use, controlling traffic, predicting the weather, and tracking the spread of diseases. One big challenge is accurately guessing what will happen far into the future based on this type of data. Such Long-term Time Series Forecasting (LTSF) is not only a research problem but also an important real-world problem.

Extensive research has been undertaken to explore the efficacy of models based on Recurrent Neural Networks (RNNs) and Transformers for addressing the challenges associated with long-term time series forecasting. Notably, Transformer models have recently achieved remarkable success in Natural Language Processing (NLP) and Computer Vision (CV) Vaswani et al. (2017); Devlin et al. (2018); Dosovitskiy et al. (2020). This success is due to the fact that the self-attention mechanism based on query key interaction appropriately accommodates short-range and long-range dependencies. Given their exceptional capabilities for sequential data processing, Transformer-based models (Zhou et al., 2021; Liu et al., 2021a; Kitaev et al., 2020) are being extensively studied for their applicability to time series forecasting, with particular focus on the complex challenges posed by LTSF.

In recent years, the application of Transformer-based models has marked a significant breakthrough in the field of long-term time series forecasting. These models have substantially leveraged the understanding and decoding of intricate temporal dependencies and patterns present in long-term time-series data. Despite their notable advancements, they harbor intrinsic limitations that preclude optimal functionality. One primary limitation is the challenge posed by the direct extraction of temporal dependencies from long-term data sequences, a process often complicated by the intricate temporal patterns muddling these dependencies Wu et al. (2021). Furthermore, the computational burden associated with Transformers, which exponentially increases with sequence length due to their reliance on self-attention mechanisms. This restricts long-term predictions, marking them as computationally unsustainable given the quadratic complexity involved.

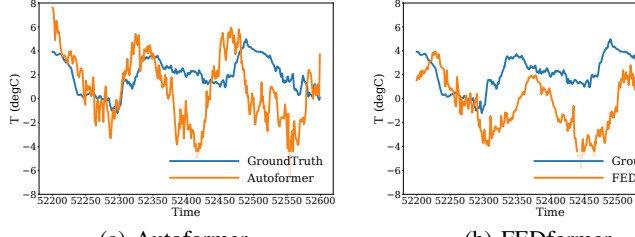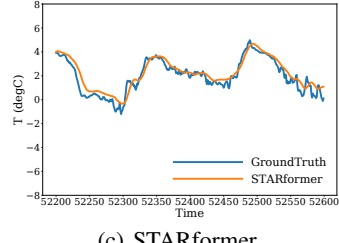

(a) Autoformer           (b) FEDformer          (c) STARformer

Figure 1: Visualization of forecasting results on Weather dataset. Blue lines are the ground truth and orange lines are the model prediction. STARformer can capture the trend and predict close to the ground truth.

To address these limitations, recent research endeavors have predominantly focused on optimizing the self-attention mechanism through the introduction of sparse variants, thereby attempting to streamline computational demands and enhance performance Zhou et al. (2021); Liu et al. (2021a). Despite these efforts, a significant shortfall persists as the revised self-attention mechanism inadequately captures the global view of time series, thereby leaving room for further enhancements.

To address this issue, several recent efforts have been aimed at reorganizing the self-attention mechanism around alternative approaches pioneered by Autoformer and FEDformer. A pivotal advancement in this area is the new architectural innovation of Autoformer, which, for the first time, replaces the traditional self-attention mechanism with an auto-correlation mechanism. Enhanced with progressive decomposition capabilities, this reimagined framework has demonstrated outstanding performance, especially in managing complex time series data Wu et al. (2021). At the same time, FEDformer adopts a harmonious approach, combining Transformer with seasonal trend decomposition and Fourier analysis to create a frequency-enhanced block that replaces the self-attention mechanism, opening a new way for the LTSF task Zhou et al. (2022b).

Our work aligns well with recent research trends. Our method extracts attention maps from simple linear models, which we found to be more effective than the auto-correlation and Fourier transform techniques used by Autoformer and FEDformer. Previous works, i.e., FEDformer and Autoformer, show that attention maps based on the characteristics of time series is more effective than the self-attention method. In this paper, we decompose complex time series into trend and seasonal patterns, subsequently learning these trend and seasonal patterns through a pre-trained single-linear layer model. The predicted trend and seasonal patterns offer valuable insights for creating attention maps. Figure 1 shows time series forecasting results on the Weather dataset, where Autoformer and FEDformer well predict seasonal part in time series but fail for trends. However, when compared to them, our model, STARformer, predicts all the details of the time series. We conduct experiments on 9 real-world datasets in energy consumption, traffic and economics planning, and weather and disease propagation forecasting. We compare our method with Transformer-based baselines and other types of baselines. Our method outperforms them in all cases. Our contributions can be summarized as follows:

1. We propose **ST**rucrural **A**ttention t**R**ans**former**, called STARformer, to deal with complex temporal patterns in the long-term time series. In this paper, we decompose complex time series into simple time series to make forecasting problems easily solved.

2. Structural attention is generated based on trends predicted by a pre-trained single-layer model. Since it uses predicted trends, it contains insightful information about where to focus in the input sequence.

3. Since our model, STARformer, replaces the self-attention with our structural attention extracted from a pre-trained model, the time complexity is greatly reduced compared to many existing models.

4. We conduct extensive experiments over 9 benchmark datasets across multiple domains. Our empirical studies show that the proposed method, called STARformer, significantly outperforms existing 12 baselines.

## 2 RELATED WORK

**Transformer for time-series forecasting** Since its introduction, Transformer models have been widely adopted for a variety of natural language tasks and computer vision tasks. However, for long-term time series forecasting tasks, conventional transformers have some limitations. There are two limitations: i) increased computational complexity for extensive calculations, and ii) inability to capture global trends. Research in fields like computer vision and natural language processing is actively addressing these challenges, spurring renewed interest in studying Transformer-based models for time series forecasting Han et al. (2022); Liu et al. (2021b); Khan et al. (2022); Wolf et al. (2020); Kalyan et al. (2021). Consequently, there has been a renewed enthusiasm in the scholarly community to deepen the study of Transformer-based models for time-series forecasting.

In this landscape, LogTrans Li et al. (2019), Informer Zhou et al. (2021), and Pyraformer Liu et al. (2021a) tackle attention mechanism complexities. LogTrans uses LogSparse attention to reduce complexity to $(O(LlogL)$. Informer applies ProbSparse attention, achieving similar efficiency, while Pyraformer implements pyramidal attention for linear complexity. PatchTST Nie et al. (2022) improves this by segmenting time series before using a Transformer, showing better performance against existing models. Despite being anchored in the underlying Transformer architecture, innovations aim to transition from self-attention to sparse self-attention, often missing a global time series view. Recently, rather than maintaining the self-attention mechanism that exists in Transformers, research is being conducted in the direction of integrating prior knowledge about the time series structure, such as Autoformer Wu et al. (2021) and ETSformer Woo et al. (2022). Autoformer uses an auto-correlation attention for periodic patterns but lacks in series decomposition, overly relying on a basic moving average for detrending, which may limit capturing complex trend patterns. ETSformer employs exponential smoothing in Transformers and introduces exponential smooth attention (ESA) and frequency attention (FA) mechanisms to replace standard self-attention, aiming for accuracy, efficiency, and interpretable decomposition. FEDformer Zhou et al. (2022b) integrates Transformer with seasonal trend decomposition, leveraging decomposition for global profiles and Transformers for detailed structures. For enhanced long-term forecasting, FEDformer uses sparse time series representations like Fourier transforms to propose a frequency-enhanced Transformer.

In this paper, we aim at a methodology for deriving attention maps utilizing simple linear models, based on recent advances in the field, represented by works such as Autoformer and ETSformer. The core of our method is to decompose complex time series into identifiable trends and seasonal fluctuations. This process is facilitated by a pre-trained single linear layer model. These approaches offer promising avenues for more informed and agile time series analysis by promoting a nuanced understanding of evolving patterns and directing attention based on rich insights derived from expected trends and seasonal dynamics.

Table 1: Comparison of attention types in recent Transformers models

| Models | Key-query based attention | Frequency based attention | Decomposition based attention |
|---|---|---|---|
| Pyraformer | O | X | X |
| Informer | O | X | X |
| Autoformer | X | X | O |
| ETSformer | X | X | O |
| PatchTST | O | X | X |
| FEDformer | X | O | O |
| STARformer | X | X | O |

**Attention mechanism in time-series** Transformers based on the self-attention mechanism show great power in sequential data, such as NLP, audio processing and even CV Vaswani et al. (2017); Wu et al. (2020). In long-term time series forecasting, The self-attention in transformer is computationally intensive and struggles to capture overall trends. Efforts to mitigate these issues have led to the development of models such as Informer and Pyraformer, which adopt a sparse rendition of the self-attention mechanism, aiming to reduce the computational burden traditionally associated with Transformers. However, it should be noted that these endeavors principally maintain a point-wise dependency and aggregation approach. Recent research approaches (cf.Table. 1), such as Autoformer and ETSformer, proposed the existing point-wise dependency attention as a new series-wise attention and a new transformer architecture that reflects the characteristics of time series (e.g., trend, seasonal part, and Fourier transform). This research stream has led to significant improvements, including performance improvements of more than 30% in long-term time series forecasting tasks. It is within this context that our study situates itself, aiming to further this positive trend in research by exploring innovative strategies to boost both computation and performance in long-term time series forecasting.

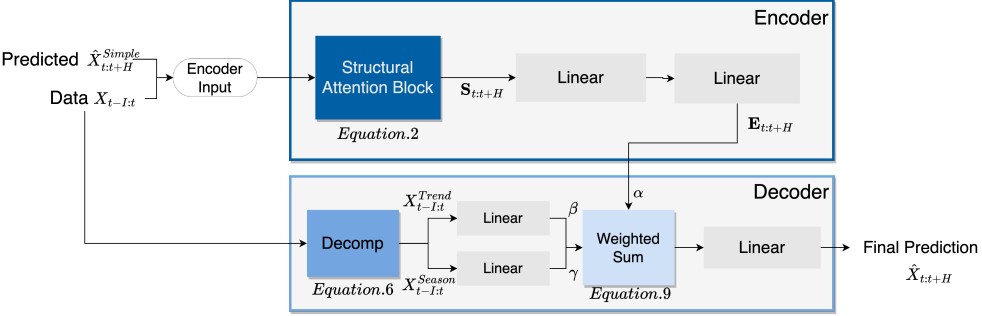

Figure 2: Overall Architecture

**Time-series decomposition**  Time series decomposition Lv & Wang (2022); Abdollahi (2020) is a well-established technique aimed at breaking down a time series into its underlying components, primarily trend, seasonality, and residuals. This decomposition facilitates a deeper understanding of the patterns and structures present in the data, aiding in more accurate and interpretable forecasting. This technique forms an essential pre-processing step in forecasting efforts but finds application in algorithms such as Prophet, which uses trend-seasonal decomposition, N-BEATS Oreshkin et al. (2019) and N-Hits Challu et al. (2023), which exploit basis extensions, and DeepGLO Sen et al. (2019) with matrix decomposition. The decomposition approach – not without limitations Wang et al. (2023) shows performance improvement in time series forecasting by decomposing a complex time series into trend and seasonal parts and then learning each decomposed time series through various modules. Primarily, this results from a rather surface-level analysis of historical data series, which tends to bypass the complex hierarchical interrelationships between the various series patterns projected on extended future timelines. STARformer addresses time-series decomposition limitations with a single linear layer model that learns future trends and seasonal parts from past data.

## 3 STARFORMER

In this section, we propose STARformer, a new architecture that replaces the self-attention of the existing Transformer architecture with structural attention generated based on the characteristics of time series. We will introduce (1) how to generate structural attention, (2) The overall architecture of STARformer as shown in Figure 2, and (3) training algorithms.

### 3.1 HOW TO GENERATE STRUCTURAL ATTENTION

One inherent drawback of the key-query-based self-attention mechanism is its difficulty to derive long-term predictions through the direct extraction of temporal dependencies from long sequences. This occurs due to the interference of complex temporal patterns that obfuscate the underlying dependencies. To mitigate this, our goal is to extract simple yet effective attention, called structural attention in our paper, relying on the decomposition of complex time series data into more rudimentary, yet informative, time series constituents such as trends, and seasonal parts. Given that linear models can effectively extract trends and seasonal parts, and their capability has been demonstrated in recent research Zeng et al. (2023), we are inspired by Zeng et al. (2023) to define our structural attention using the linear regression equation, as described in Equation 1. Additionally, Wu et al. (2021) has attempted to use predicted trends in predictions. Likewise, we ensure that our structural attention $\mathbf{S}_{t:t+H}$ is defined based on predicted simple time series (e.g., trends or seasonal parts). In Equation 1, the matrix $\mathbf{S}_{t:t+H}$ refers to the attention matrix to the input sequence $X_{t-I:t}$ for forecasting future values $X_{t:t+H}$. Suppose we have the length of input sequence $I$ and the length of forecast horizon $H$ and $d$-dimension of data $X$. For simplicity, we suppose $I = H$ (Cases, where $I$ and $H$ are different, are explained in Section 3.2).

$$X_{t:t+H} = \mathbf{S}_{t:t+H} \times X_{t-I:t} + \mathbf{B}_{t-I:t}, \tag{1}$$

where $X_{t-I:t} \in \mathbb{R}^{d \times I}$, $X_{t:t+H} \in \mathbb{R}^{d \times H}$ refers to input sequence, forecast horizon, and $\mathbf{B}_{t-I:t} \in \mathbb{R}^{d \times I}$, $\mathbf{S}_{t:t+H} \in \mathbb{R}^{d \times H}$ denotes the appropriate matrices for calculating $X_{t:t+H}$, respectively.

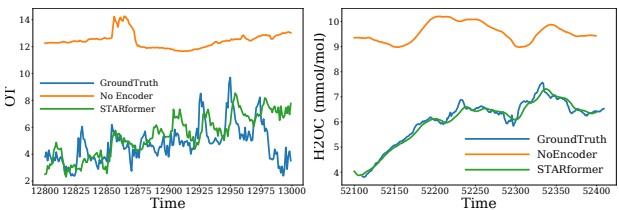

Figure 3: *Left*: ETTh1 *Right*: Weather. Compared to the case without the encoder(in orange line) and with the encoder (in green line), the encoder can match the distribution of prediction and ground truth.

The goal of Equation 1 is to derive an attention matrix, denoted as $\mathbf{S}_{t:t+H}$, which we refer to as 'structural attention' in this work. Initially, employing conventional time series decomposition techniques, the residual of $X_{t-I:t}$ can be represented as $\mathbf{B}_{t-I:t}$. Given that future data $X_{t:t+H}$ is not accessible, we opt to substitute it with simpler time series components, such as trends or seasonality. As a subsequent step, to estimate $\hat{X}_{t:t+H}^{Simple}$, we train a single-layer linear model, $f$, aimed at forecasting future trends or seasonal parts, as detailed in Algorithm 1. It's noteworthy that simpler time series components like trends or seasonal parts can be readily trained with a single linear model $f$, i.e., a fully-connected layer, enabling us to substitute $X_{t:t+H}$ with $\hat{X}_{t:t+H}^{Simple}$. Consequently, the structural attention $\mathbf{S}_{t:t+H}$ can be defined as follows:

$$\mathbf{S}_{t:t+H} = \frac{\hat{X}_{t:t+H}^{Simple} - \mathbf{B}_{t-I:t}}{X_{t-I:t}}, \qquad (2)$$

where $\hat{X}_{t:t+H}^{Simple} = f(X_{t-I:t}^{Simple}; \boldsymbol{\theta_f})$ and $\hat{X}_{t:t+H}^{Simple} \in \mathbb{R}^{d \times H}$ refers to predicted trends or seasonal parts through the single linear model $f$. So, structural attention $\mathbf{S}_{t:t+H}$ can be defined as attention that can explain future trends or seasonal parts $\hat{X}_{t:t+H}^{Simple}$ based on the input time series $X_{t-I:t}$.

---

**Algorithm 1:** How to extract structural attention

**Input:** Train input sequences $X_{Train}$ ; Validating input sequences $X_{Val}$ ; Maximum iteration number $max\_iter$.

1  Get Trend $X_{Train}^{Trend}$ and seasonal part $X_{Train}^{Season}$ from $X_{Train}$ by using `time-series decomposition method`;
2  Initialize the parameters $\boldsymbol{\theta_f}$;
3  $i \leftarrow 0$; **while** $i < max\_iter$ **do**
4      Train $\boldsymbol{\theta_f}$ and MSE loss $L_{Simple}$ for Trends (or Seasonal parts) forecasting;
5      Validate and update the best parameters $\boldsymbol{\theta_f^*}$ with $D_{val}$;
6      $i \leftarrow i + 1$;
7  **return** $\boldsymbol{\theta_f^*}$;

---

### 3.2 OVERALL ARCHITECTURE

This section outlines the overall architecture of STARformer. STARformer uses the simple encoder-decoder architecture, as shown in Figure 2, for long-term time series forecasting. Initially, it extracts the structural attention based on predicted trends or seasonal parts. Then, the encoder constructs the major forecast flow, which the decoder then refines with detailed insights. As shown in Figure 1, our model, STARformer, can predict global trends compared to the others. Finally, The final prediction $\hat{X}_{t:t+H}$ can be defined as follows:

$$\hat{X}_{t:t+H} = Decoder(\mathbf{E}_{t:t+H}, X_{t-I:t}), \qquad (3)$$

where $\mathbf{E}_{t:t+H} \in \mathbb{R}^{d \times H}$ and $X_{t-I:t} \in \mathbb{R}^{d \times I}$ refers to the encoder output, and input sequence.

**Encoder** The encoder focuses on structural attention. Through structural attention in STARformer, we can i) solve the distribution discrepancy problem, and ii) construct the major forecast flow. First, the distribution discrepancy between ground-truth and prediction results is known to be a limitation of the self-attention of conventional transformers Zhou et al. (2022a). Zhou et al. (2022b) solves the distribution discrepancy issue by replacing self-attention. Likewise, STARformer overcomes distribution discrepancy by replacing self-attention with structural attention. Second, since the structural attention is generated based on the predicted $\hat{X}_{t:t+H}^{Simple}$ as in Equation (4) (cf.See Subsection 3.1), it has a great advantage in predicting a major forecast flow of data. As shown in

Figure 3, it clearly shows the role of the encoder's structural attention by comparing the case without an encoder (orange line) and the case with an encoder (green line). Structural attention $\mathbf{S}_{t:t+H}$ ,which is the core part of the encoder, is decided according to the length of input sequence $I$ and forecast horizon $H$, so our structural attention $\mathbf{S}_{t:t+H}$ created from the predicted $\hat{X}_{t:t+H}^{Simple}$ can be defined as follows:

$$
\begin{aligned}
\mathbf{S}_{t:t+I} &= \frac{\hat{X}_{t:t+I}^{Simple} - \mathbf{B}_{t-I:t}}{X_{t-I:t}}, \qquad \text{if } I \leq H \\
\mathbf{S}_{t:t+H} &= \frac{\hat{X}_{t:t+H}^{Simple} - \mathbf{B}_{t-H:t}}{X_{t-H:t}}. \qquad \text{if } I > H
\end{aligned}
\tag{4}
$$

Finally, our encoder can be defined as follows:

$$
\mathbf{E}_{t:t+H} = \begin{cases} Linear_{I \to H}(Linear_{I \to I}(\mathbf{S}_{t:t+I} \times X_{t-I:t})), & \text{if } I \leq H \\ Linear_{H \to H}(Linear_{H \to H}(\mathbf{S}_{t:t+H} \times X_{t-H:t})). & \text{if } I > H \end{cases}
\tag{5}
$$

Note that linear layer $Linear_{I \to H}: \mathbb{R}^I \to \mathbb{R}^H$ and $Linear_{H \to H}: \mathbb{R}^H \to \mathbb{R}^H$.

**Decoder** The decoder focuses on analyzing the input sequence $X_{t-I:t}$ and feedforward operations for the final prediction. Input sequence analysis involves i) converting the input sequence $X_{t-I:t}$ to trends $X_{t-I:t}^{Trend}$ and seasonal parts $X_{t-I:t}^{Season}$ (cf. Equation 6). ii) Each feature extractor $g, h$ is used to model the main features of the trends and seasonal parts. iii) The final prediction is the weighted sum of the encoder output $\mathbf{E}_{t:t+H}$, trends $X_{t-I:t}^{Trend}$ and seasonal parts $X^{Season}$. The encoder focuses on modeling future trends (or seasonal parts) through structural attention, while the decoder focuses on modeling detailed points of the input sequence $X_{t-I:t}$. Especially in time series forecasting tasks, being able to add detailed points is a big part of evaluating model performance. The sequential approach of STARformer helps easily solve complex long-term time series forecasting tasks.

$$
X_{t-I:t}^{Trend}, X_{t-I:t}^{Season} = \text{Decomposition}(X_{t-I:t}),
\tag{6}
$$

$$
X_{t-I:t}^{*Trend} = Linear_{I \to H}(X_{t-I:t}^{Trend}),
\tag{7}
$$

$$
X_{t-I:t}^{*Season} = Linear_{I \to H}(X_{t-I:t}^{Season}),
\tag{8}
$$

$$
\mathbf{D}_{t:t+H} = \alpha \times \mathbf{E}_{t:t+H} + \beta \times X_{t-I:t}^{*Trend} + \gamma \times X_{t-I:t}^{*Season},
\tag{9}
$$

$$
\hat{X}_{t:t+H} = Linear_{H \to H}(\mathbf{D}_{t:t+H}),
\tag{10}
$$

where $\{X_{t-I:t}^{Trend}, X_{t-I:t}^{Season}\} \in \mathbb{R}^{d \times I}$ refers to trends and seasonal parts from the input sequence $X_{t-I:t}$ and $\{X_{t-I:t}^{*Trend}, X_{t-I:t}^{*Season}\} \in \mathbb{R}^{d \times H}$ are hidden representation of trends and seasonal parts. $\alpha, \beta$, and $\gamma$ are the coefficients for the decoder output $\mathbf{D}_{t:t+H} \in \mathbb{R}^{d \times H}$ in Equation 9.

### 3.3 TRAINING METHOD

Our method, STARformer, has two training steps. First, the process of learning a single linear layer to generate structured attention (cf. Algorithm 1). Second, the training process for long-term time series forecasting. In this step, STARformer learns using structured attention extracted from the pre-trained Algorithm 1. We use the mean squared error (MSE) loss for forecasting. Instead of the existing 2D loss curve, we compare the learning method of our model with that of the existing model through loss landscape. Loss landscape Li et al. (2018); Park & Kim (2022) is a 3D visualization of the loss value that changes depending on the perturbation given to the weights of the neural network model. In Figure 4, we visualize the loss landscape for STARformer and 3 other transformer-based models. Compared to other methods, STARformer has a flatter loss landscape than Informer near the optimum. Autoformer and Pyraformer, they reach an optimally flat loss landscape, while STARformer has an almost perfectly smooth parabolic loss landscape, leading to better neural network optimization.

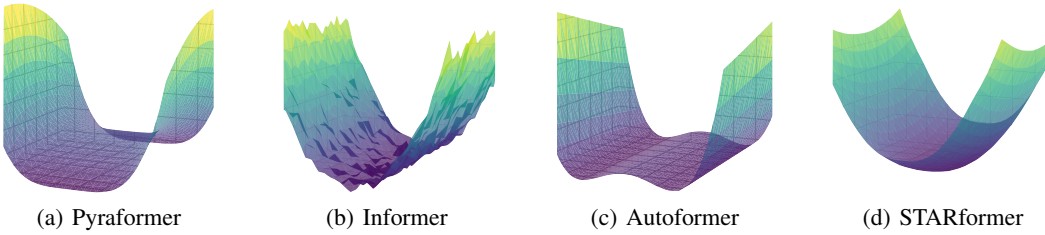

| (a) Pyraformer | (b) Informer | (c) Autoformer | (d) STARformer |

Figure 4: Loss landscape visualizations on the national illness dataset.

Table 2: Experimental results on 5 benchmarked dataset. We use forecast horizons $H \in \{24, 36, 48, 60\}$ for the national illness and $H \in \{96, 192, 336, 720\}$ for the others. The best results are in **bold** and the second best are underlined.

| Datasets | horizons | STARformer MSE | STARformer MAE | ETSformer MSE | ETSformer MAE | PatchTST MSE | PatchTST MAE | PatchTST(-in) MSE | PatchTST(-in) MAE | FEDformer MSE | FEDformer MAE | Autoformer MSE | Autoformer MAE | Informer MSE | Informer MAE | Pyraformer MSE | Pyraformer MAE | LogTrans MSE | LogTrans MAE |
|---|---|---|---|---|---|---|---|---|---|---|---|---|---|---|---|---|---|---|---|
| Electricity | 96 | **0.084** | **0.198** | 0.187 | 0.304 | 0.129 | 0.222 | 0.413 | 0.295 | 0.186 | 0.302 | 0.201 | 0.317 | 0.274 | 0.368 | 0.386 | 0.449 | 0.258 | 0.357 |
| | 192 | **0.070** | **0.178** | 0.199 | 0.315 | 0.147 | 0.240 | 0.425 | 0.302 | 0.201 | 0.315 | 0.222 | 0.334 | 0.296 | 0.386 | 0.386 | 0.443 | 0.266 | 0.368 |
| | 336 | **0.076** | **0.194** | 0.212 | 0.329 | 0.163 | 0.259 | 0.435 | 0.307 | 0.214 | 0.329 | 0.231 | 0.338 | 0.300 | 0.394 | 0.378 | 0.443 | 0.280 | 0.380 |
| | 720 | **0.069** | **0.177** | 0.233 | 0.345 | 0.197 | 0.290 | 0.473 | 0.321 | 0.246 | 0.355 | 0.254 | 0.361 | 0.373 | 0.439 | 0.376 | 0.445 | 0.283 | 0.376 |
| Traffic | 96 | **0.335** | **0.299** | 0.607 | 0.392 | 0.360 | 0.249 | 0.413 | 0.295 | 0.587 | 0.366 | 0.613 | 0.388 | 0.719 | 0.391 | 2.085 | 0.468 | 0.684 | 0.384 |
| | 192 | **0.278** | **0.269** | 0.621 | 0.399 | 0.379 | 0.256 | 0.425 | 0.302 | 0.604 | 0.373 | 0.616 | 0.382 | 0.696 | 0.379 | 0.867 | 0.467 | 0.685 | 0.390 |
| | 336 | **0.263** | **0.258** | 0.622 | 0.396 | 0.392 | 0.264 | 0.435 | 0.307 | 0.621 | 0.383 | 0.622 | 0.338 | 0.777 | 0.420 | 0.869 | 0.469 | 0.734 | 0.408 |
| | 720 | **0.259** | **0.253** | 0.632 | 0.396 | 0.432 | 0.286 | 0.473 | 0.321 | 0.626 | 0.382 | 0.660 | 0.408 | 0.864 | 0.472 | 0.881 | 0.473 | 0.717 | 0.396 |
| Weather | 96 | **0.050** | **0.084** | 0.197 | 0.281 | 0.149 | 0.198 | 0.161 | 0.219 | 0.217 | 0.296 | 0.266 | 0.336 | 0.300 | 0.384 | 0.896 | 0.556 | 0.458 | 0.490 |
| | 192 | **0.050** | **0.084** | 0.237 | 0.312 | 0.194 | 0.241 | 0.201 | 0.254 | 0.276 | 0.336 | 0.307 | 0.367 | 0.598 | 0.544 | 0.622 | 0.624 | 0.658 | 0.589 |
| | 336 | **0.050** | **0.084** | 0.298 | 0.353 | 0.245 | 0.282 | 0.253 | 0.298 | 0.339 | 0.380 | 0.359 | 0.395 | 0.578 | 0.523 | 0.739 | 0.753 | 0.797 | 0.652 |
| | 720 | **0.051** | **0.084** | 0.352 | 0.388 | 0.314 | 0.334 | 0.323 | 0.357 | 0.403 | 0.428 | 0.419 | 0.428 | 1.059 | 0.741 | 1.004 | 0.934 | 0.869 | 0.675 |
| ILI | 24 | **1.266** | **0.728** | 2.527 | 1.020 | 1.319 | 0.754 | 3.489 | 1.345 | 3.228 | 1.260 | 3.483 | 1.287 | 5.764 | 1.677 | 1.420 | 2.012 | 4.480 | 1.444 |
| | 36 | 1.249 | **0.729** | 2.071 | 2.615 | **1.007** | 0.870 | 3.426 | 1.205 | 2.679 | 1.080 | 3.103 | 1.148 | 4.755 | 1.467 | 7.394 | 2.031 | 4.799 | 1.467 |
| | 48 | **1.191** | **0.716** | 2.359 | 0.972 | 1.553 | 0.815 | 4.309 | 1.449 | 2.622 | 1.078 | 2.669 | 1.085 | 4.763 | 1.469 | 7.551 | 2.057 | 4.800 | 1.468 |
| | 60 | **1.492** | **0.777** | 2.137 | 2.487 | **1.016** | 0.788 | 4.065 | 1.402 | 2.857 | 1.157 | 2.770 | 1.125 | 5.264 | 1.564 | 7.662 | 2.100 | 5.278 | 1.560 |
| Exchange | 96 | **0.011** | **0.074** | 0.085 | 0.204 | 0.093 | 0.218 | 0.116 | 0.248 | 0.139 | 0.276 | 0.197 | 0.323 | 0.847 | 0.752 | 0.376 | 1.105 | 0.968 | 0.812 |
| | 192 | **0.012** | **0.075** | 0.182 | 0.303 | 0.208 | 0.332 | 0.346 | 0.440 | 0.256 | 0.369 | 0.300 | 0.369 | 1.204 | 0.895 | 1.748 | 1.151 | 1.040 | 0.851 |
| | 336 | **0.012** | **0.075** | 0.348 | 0.428 | 0.359 | 0.440 | 0.581 | 0.575 | 0.426 | 0.464 | 0.509 | 0.524 | 1.672 | 1.036 | 1.874 | 1.172 | 1.659 | 1.081 |
| | 720 | **0.013** | **0.079** | 1.025 | 0.774 | 1.194 | 0.815 | 1.604 | 0.934 | 1.090 | 0.800 | 1.447 | 0.941 | 2.478 | 1.310 | 1.943 | 1.206 | 1.941 | 1.127 |

# 4 EXPERIMENTS

In this section, we describe our experimental environments and results. We conduct experiments on LTSF. All experiments were conducted in the same software and hardware environments. UBUNTU 18.04 LTS, PYTHON 3.8.0, NUMPY 1.22.3, SCIPY 1.10.1, MATPLOTLIB 3.6.2, PYTORCH 2.0.1, CUDA 11.4, NVIDIA Driver 470.182.03 i9 CPU, and NVIDIA RTX A5000. We repeat the training and testing procedures with three different random seeds and report MSE and MAE of multivariate time series forecasting as metrics. We list all the descriptions of datasets, detailed experimental settings are in the Appendix A.

## 4.1 EXPERIMENTS RESULTS

Table 2 and Table 3 shows the experimental results on 9 benchmarked datasets. In Table 2 and Table 3, STARformer achieves the best performance with significant differences from other transformer models in all forecasting horizon lengths $H$. In particular, compared to PatchTST, which shows state-of-the-art performance, STARformer reduces the overall MSE by 46% and 51%, in Table 2 and Table 3 respectively. However, PatchTST has performance differences depending on the presence of the RevIn data normalization Kim et al. (2021) (cf. PatchTST(-in) refers to without the RevIn). Additionally, when compared to Autoformer, which proposes auto-correlation-based attention, STARformer provides an overall relative MSE reduction of 71%. We also compare with non-transformer models. When compared to models such as DLinear or FiLM, which are state-of-the-art among non-transformer-based models, our model is superior in all cases. The results of non-transformer models (NLinear, DLinear, FiLM, and N-Hits) are in the Appendix D.

Table 3: Experimental results on 4 ETT dataset.

| Datasets | | STARformer | | ETSformer | | PatchTST | | PatchTST(-in) | | FEDformer | | Autoformer | | Informer | | Pyraformer | | LogTrans | |
|---|---|---|---|---|---|---|---|---|---|---|---|---|---|---|---|---|---|---|---|
| | horizons | MSE | MAE | MSE | MAE | MSE | MAE | MSE | MAE | MSE | MAE | MSE | MAE | MSE | MAE | MSE | MAE | MSE | MAE |
| ETTh1 | 96 | **0.218** | **0.193** | 0.507 | 0.484 | 0.370 | 0.400 | 0.385 | 0.410 | 0.376 | 0.419 | 0.449 | 0.459 | 0.865 | 0.713 | 0.664 | 0.612 | 0.878 | 0.740 |
| | 192 | **0.211** | **0.199** | 0.554 | 0.509 | 0.413 | 0.429 | 0.417 | 0.432 | 0.420 | 0.448 | 0.500 | 0.482 | 1.008 | 0.792 | 0.790 | 0.681 | 1.037 | 0.824 |
| | 336 | **0.203** | **0.223** | 0.591 | 0.526 | 0.422 | 0.440 | 0.439 | 0.449 | 0.459 | 0.465 | 0.521 | 0.496 | 1.107 | 0.809 | 0.891 | 0.738 | 1.238 | 0.932 |
| | 720 | **0.296** | **0.364** | 0.581 | 0.538 | 0.447 | 0.468 | 0.478 | 0.494 | 0.506 | 0.507 | 0.514 | 0.512 | 1.181 | 0.865 | 0.963 | 0.782 | 1.135 | 0.852 |
| ETTh2 | 96 | **0.127** | **0.151** | 0.345 | 0.399 | 0.274 | 0.336 | 0.299 | 0.359 | 0.346 | 0.388 | 0.358 | 0.397 | 3.755 | 1.525 | 0.645 | 0.597 | 2.116 | 1.197 |
| | 192 | **0.137** | **0.190** | 0.434 | 0.445 | 0.339 | 0.379 | 0.354 | 0.404 | 0.429 | 0.439 | 0.456 | 0.452 | 5.602 | 1.931 | 0.788 | 0.683 | 4.315 | 1.635 |
| | 336 | **0.164** | **0.250** | 0.410 | 0.447 | 0.329 | 0.384 | 0.374 | 0.420 | 0.496 | 0.487 | 0.482 | 0.486 | 4.721 | 1.835 | 0.907 | 0.747 | 1.124 | 1.604 |
| | 720 | **0.284** | **0.371** | 0.475 | 0.486 | 0.379 | 0.422 | 0.479 | 0.492 | 0.463 | 0.474 | 0.515 | 0.511 | 3.647 | 1.625 | 0.963 | 0.783 | 3.188 | 1.540 |
| ETTm1 | 96 | **0.116** | **0.228** | 0.373 | 0.396 | 0.290 | 0.342 | 0.308 | 0.358 | 0.379 | 0.419 | 0.505 | 0.475 | 0.672 | 0.571 | 0.543 | 0.510 | 0.600 | 0.546 |
| | 192 | **0.101** | **0.211** | 0.404 | 0.407 | 0.332 | 0.369 | 0.335 | 0.375 | 0.426 | 0.441 | 0.553 | 0.496 | 0.795 | 0.669 | 0.557 | 0.537 | 0.837 | 0.700 |
| | 336 | **0.098** | **0.210** | 0.431 | 0.424 | 0.366 | 0.392 | 0.362 | 0.392 | 0.445 | 0.459 | 0.621 | 0.537 | 1.212 | 0.871 | 0.754 | 0.655 | 1.124 | 0.832 |
| | 720 | **0.098** | **0.211** | 0.494 | 0.456 | 0.416 | 0.420 | 0.432 | 0.429 | 0.543 | 0.490 | 0.671 | 0.561 | 1.166 | 0.823 | 0.908 | 0.724 | 1.153 | 0.820 |
| ETTm2 | 96 | **0.137** | **0.135** | 0.189 | 0.280 | 0.165 | 0.255 | 0.167 | 0.257 | 0.203 | 0.287 | 0.255 | 0.339 | 0.365 | 0.453 | 0.435 | 0.507 | 0.768 | 0.642 |
| | 192 | **0.136** | **0.135** | 0.253 | 0.319 | 0.220 | 0.292 | 0.226 | 0.303 | 0.269 | 0.328 | 0.281 | 0.340 | 0.533 | 0.563 | 0.730 | 0.673 | 0.989 | 0.757 |
| | 336 | **0.133** | **0.135** | 0.314 | 0.357 | 0.274 | 0.329 | 0.301 | 0.348 | 0.325 | 0.366 | 0.339 | 0.372 | 1.363 | 0.887 | 1.201 | 0.845 | 1.334 | 0.872 |
| | 720 | **0.132** | **0.157** | 0.414 | 0.413 | 0.362 | 0.385 | 0.392 | 0.407 | 0.421 | 0.415 | 0.433 | 0.432 | 3.379 | 1.338 | 3.625 | 1.451 | 3.048 | 1.328 |

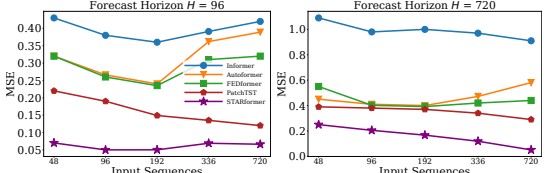

(a) Dataset      (b) Autoformer      (c) STARformer

Figure 5: (a): Visualization of 5 columns of national illness data (% Weighted ILI, %Unweighted ILI, AGE0-4, AGE5-24, ILITOTAL from the top) (b): Visualization of autoformer's attention for 5 columns (c): Visualization of starformer's attention for 5 columns

## 4.2 VISUALIZATION ON ATTENTION

In the national disease dataset in Figure 5(a), an increasing trend and repeating increases and decreases over time are observed in all five columns visualized. In this subsection, we analyze and compare the attention visualizations of autoformer and starformer. Since the autoformer has an auto-correlation-based structure, the attention from autoformer shows repeated seasonality for each column. On the other hand, our model uses structural attention, which generates based on trends predicted by a single-linear model. Therefore, it appears to reflect seasonality well, and although the attention intensity for each column is different, structural attention reflects an increasing trend in most cases. We additionaly visualize the actual data and attention maps in the Appendix B.2.

## 4.3 SENSITIVITY ANALYSIS & ABLATION STUDY

**Varying input sequence length** In theory, longer input sequences provide more information for the model to learn, potentially improving prediction accuracy. However, this notion is refuted by the results presented in Zeng et al. (2023), which show the lack of this improvement in most Transformer-based models. In Figure 6, we measure the MSE in experiments conducted with various input sequences. As the input sequence is longer, Transformer-based models show limited performance. STARformer shows excellent performance regardless of the input sequence., and the MSE decreases as the input sequence becomes longer. Visualization results on other datasets are in the Appendix B.3.

Figure 6: *Left*: Forecast Horizon $H = 96$ *Right*: $H = 720$. Forecasting performance (MSE) with varying input sequences on the Weather dataset. Input sequence is $\{48, 96, 192, 336, 720\}$ and Forecast horizon is $\{96, 720\}$.

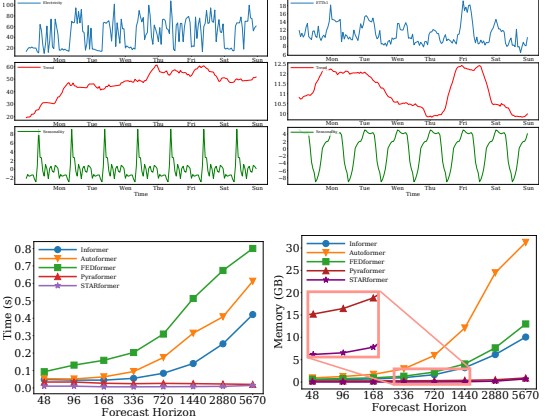

Figure 7: *Left*: Electricity *Right*: ETTh1. From the top, we visualize the actual data, trends, and seasonal parts of the data.

Figure 8: *Left*: Runtime efficiency *Right*: Memory efficiency. We measured runtime and memory efficiency on the ETTm2 dataset with various forecast horizons $H \in \{48, 96, 192, 336, 720\}$.

**Extract structural attention from seasonal part** Time series data consists of trends and seasonal parts, which differ based on the dataset. For instance, the ETT dataset has trends, whereas the Electricity has seasonal elements. STARformer utilizes these components to generate attention. The success of the structural attention in our model is influenced by which component represents the data, highlighting the importance of choosing the right attention type for effective LTSF problem-solving. Table 4 summarizes the results of ablation studies for the electricity and ETTh1 datasets. Figure 7 displays one week of Electricity, showing seasonality that is as meaningful as trends. Regarding the results on Electricity of Table 4, the performance of STARformers — season and trend— outperforms PatchTST, which is state-ot-the-art of transformer based models. On the other hand, for ETTh1, where trends are more meaningful than seasonal parts, such as Fig-

Table 4: Ablation study on extracting structural attention from season

| Datasets | | Electricity | | ETTh1 | |
|---|---|---|---|---|---|
| | horizons | MSE | MAE | MSE | MAE |
| PatchTST | 96 | 0.129 | 0.222 | 0.370 | 0.400 |
| | 192 | 0.147 | 0.240 | 0.413 | 0.429 |
| | 336 | 0.163 | 0.259 | 0.422 | 0.440 |
| | 720 | 0.197 | 0.290 | 0.447 | 0.468 |
| STARformer season | 96 | 0.110 | 0.188 | 0.359 | 0.387 |
| | 192 | 0.092 | **0.172** | 0.403 | 0.387 |
| | 336 | 0.095 | **0.177** | 0.432 | 0.390 |
| | 720 | 0.113 | 0.194 | 0.451 | 0.429 |
| STARformer trend | 96 | **0.084** | 0.198 | **0.218** | 0.193 |
| | 192 | **0.070** | 0.178 | **0.211** | **0.199** |
| | 336 | **0.076** | 0.194 | **0.203** | **0.223** |
| | 720 | **0.069** | 0.177 | **0.296** | **0.364** |

ure 7, STARformer (trend) performance is better than STARformer (season). The experimental results on structural attention with seasonal parts on the other 7 datasets are in the Appendix C.

## 4.4 COMPLEXITY ANALYSIS

STARformer has an advantage in terms of complexity by replacing existing self-attention with simple structured attention. The proposed STARformer achieves better long-term sequence efficiency by demonstrating $\mathcal{O}(L)$ and $\mathcal{O}(1)$ times in memory. Figure 8 visualize the experimental results of the training phase of ETTm2. The input order is fixed to 48, and the forecast horizon is varied from $\{48, 96, 468, 336, 720, 1440, 2880, 5670\}$. As shown in Figure 8, the time and memory cost of the proposed STARformer is approximately a linear function of $L$, as expected, and is the least memory and time complexity among the Transformer-based models.

## 5 CONCLUSION

This paper proposes a structural attention transformer model for long-term time series forecasting which achieves state-of-the-art performance. To deal with complex temporal patterns in the long-term time series, we propose Structural attention. Our model extracts i) from a single linear layer model aimed at predicting simple time series (Trend, Seasonal parts) to easily solve complex time series, and ii) structural interest based on future trends (or seasonal parts). Since our model, STARformer, replaces self-attention, time complexity and computation are greatly reduced compared to many existing models. STARformer outperforms in most cases when compared to 9 benchmark datasets and 12 baselines across multiple domains.

**Reproducibility Statement** To ensure the reproducibility and completeness of this paper, we make our code available at https://drive.google.com/drive/folders/1sfnDxYHs2i07RuflN_1K6tUVy9SwJSkG?usp=sharing. We give details on our experimental protocol in the Appendix A. Appendix A.3 contains the detailed parameters to create trends and seasonal parts from the datasets. Appendix A.4 provides the parameter to train a single linear layer to predict future trends or seasonal parts. Finally, Appendix A.5 provides the parameter to train STARformer.

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
