## A    EXPERIMENTAL DETAILS

### A.1    DATASETS

We evaluate the performance of the proposed STARformer on six benchmarking datasets, including electricity, transportation, weather, national weather, Exchange, and four ETT datasets (ETTh1, ETTh2, ETTm1, ETTm2). These benchmarked datasets have been utilized extensively and publicly available at Wu et al. (2021). The following is a description of the six experimental datasets. (1) **Electricty** contains electricity consumption per person for 321 people from 2012 to 2014. (2) **Traffic** is hourly road occupancy data collected by various sensors from the California Department of Transportation. (3) **Weather** is data measuring 21 weather indicators such as temperature, humidity, etc. every 10 minutes throughout 2020. (4) **ILI** contains patient data from the U.S. Centers for Disease Control and Prevention for weekly recorded influenza-like illness (ILI) from 2002 to 2021, describing the proportion of patients observed with ILI and the total number of patients. (5) **Exchange** contains exchange data from 8 countries Lai et al. (2018). (6) There are four types of **ETT** data sets Zhou et al. (2021). ETTh1 and ETTh2 were measured every hour, and ETTm1 and ETTm2 were recorded every 15 minutes. All four datasets were measured between July 2016 and July 2018 and contain data collected from electrical transformers, including load and oil temperature.

### A.2    BASELINES

We choose the state-of-the-art transformer-based models, including PatchTST Nie et al. (2022), FEDformer Zhou et al. (2022b), Autoformer Wu et al. (2021), ETSformer Woo et al. (2022), Informer Zhou et al. (2021), Pyraformer Liu et al. (2021a), logTrans Li et al. (2019), and a recent non-transformer-based model N-Hits Challu et al. (2023), FiLM Zhou et al. (2022a), DLinear Zeng et al. (2023), NLinear as our baselines. All of the models follow the same experimental setup with prediction horizon $H \in \{24, 36, 48, 60\}$ for national illness dataset and $H \in \{96, 192, 336, 720\}$ for other datasets as in the other papers. Since there exists a debate on the effectiveness of input sequence length, we run our model and other baselines for different input sequence length $I \in \{24, 36, 48, 60, 96\}$ for national illness dataset and $I \in \{48, 96, 192, 336, 720\}$ for other datasets. We choose the best results to create strong baselines.

### A.3    EXTRACT TRENDS OR SEASONAL PARTS FROM DECOMPOSITION METHOD

Since Structural Attention is obtained based on predicted trends or seasonal parts, we need Trend and Seasonal parts to be used for learning. To calculate this, we use the decomposition block provided by Autoformer. The kernel size and period for calculating the trend and seasonal part follow Table 5 for each dataset.

Table 5: Hyperparameter to extract trends or seasonal parts from decomposition method

| Datasets | Kernel size | Period |
|---|---|---|
| Electricity | 6 | 24 |
| Traffic | 6 | 48 |
| Weather | 25 | 24 |
| Exchange rate | 25 | 24 |
| ETT-datsets | 25 | 48 |
| ILI | 25 | 24 |

### A.4    TRAIN SINGLE LINEAR LAYER MODEL FOR STRUCTURAL ATTENTION

We train a single-linear layer model to predict future trends or seasonal parts and the following parameters as shows in Table 6.

### A.5    TRAIN STARFORMER

STARformer uses the simple encoder-decoder architecture. As shown in Figure 2, the encoder consists of structural attention blocks and linear layers. As described in Section 3, our model consists of linear layers, and the parameters vary depending on the input sequence $I$ and forecast horizon $H$. The experiments in Table 2, Table 3 and Table 8 are conducted by setting the input sequence $I$ and forecast horizon $H$ to be the same, and also the experiment can be performed when the input sequence $I$ and forecast horizon $H$ are different. For the decoder's decomposition block, the moving average kernel size is 25, which is the same as Autoformer. Algorithm 2 shows how to train STARformer.

Table 6: Hyperparameter to train single linear layer for structural attention

| Datasets | | Input Sequence | Forecast Horizon | Learning Rate | Batch size |
|---|---|---|---|---|---|
| Electricity | Trend | 96 | 12 | 0.001 | 16 |
| | Seasonality | 96 | 12 | 0.001 | 16 |
| Traffic | Trend | 336 | 6 | 0.05 | 16 |
| | Seasonality | 96 | 12 | 0.05 | 16 |
| Weather | Trend | 336 | 24 | 0.0001 | 16 |
| | Seasonality | 96 | 12 | 0.0001 | 16 |
| Exchange Rate | Trend | 96 | 12 | 0.0005 | 8 |
| | Seasonality | 96 | 12 | 0.0005 | 8 |
| ETTh1 | Trend | 96 | 12 | 0.005 | 32 |
| | Seasonality | 96 | 24 | 0.005 | 32 |
| ETTh2 | Trend | 96 | 12 | 0.05 | 32 |
| | Seasonality | 96 | 12 | 0.05 | 32 |
| ETTm1 | Trend | 336 | 6 | 0.0001 | 32 |
| | Seasonality | 96 | 12 | 0.0001 | 32 |
| ETTm2 | Trend | 336 | 6 | 0.001 | 32 |
| | Seasonality | 96 | 96 | 0.001 | 32 |
| ILI | Trend | 104 | 24 | 0.0001 | 32 |
| | Seasonality | 144 | 24 | 0.0001 | 32 |

1. For Electricity, we use learning rate 0.001, batch size = 16, epochs = 30.

2. For Weather, we use learning rate 0.0001, batch size = 16, epochs = 10.

3. For Exchange rate, we use learning rate 0.0005, batch size = 8, epochs = 10.

4. For Traffic, we use learning rate 0.001, batch size = 16, epochs = 10.

5. For National illness, we use learning rate 0.05, batch size = 32, epochs = 40.

6. For ETTh1, we use learning rate 0.005, batch size = 32, epochs = 10.

7. For ETTh2, we use learning rate 0.005, batch size = 32, epochs = 10.

8. For ETTm1, we use learning rate 0.001, batch size = 32, epochs = 10.

9. For ETTm2, we use learning rate 0.001, batch size = 32, epochs = 10.

---

**Algorithm 2:** How to train STARformer

---

**Input:** Train input sequences $X_{Train}$ ; Validating input sequences $X_{Val}$ ; Maximum iteration number $max\_iter$.

1   Initialize the parameters $\boldsymbol{\theta}$ ;

2   $i \leftarrow 0$; **while** $i < max\_iter$ **do**

3      Train $\boldsymbol{\theta}$ and MSE loss $L$ for forecasting;

4      Validate and update the best parameters $\boldsymbol{\theta}^*$ with $D_{val}$;

5      $i \leftarrow i + 1$;

6   **return** $\boldsymbol{\theta}^*$;

---

## B   VISUALIZATIONS

### B.1   VISUALIZATION FORECASTING RESULTS FROM STARFORMER ON ALL 9 BENCHMARKED DATASETS

We visualize forecasting results on all 9 benchmarked datsets. We fixed input sequence length and forecast horizon to 36 for national illness and 336 for other datasets. As shown in Figure 9, STARformer can predict the overall trends and fluctuates of time series data.

### B.2   VISUALIZATION ON ATTENTION HEATMAP

We visualize the 5 columns of each dataset and the attention to these 5 columns. Most visualized data shows repeated increases or decreases or increasing (or decreasing) trends. Among the visualized

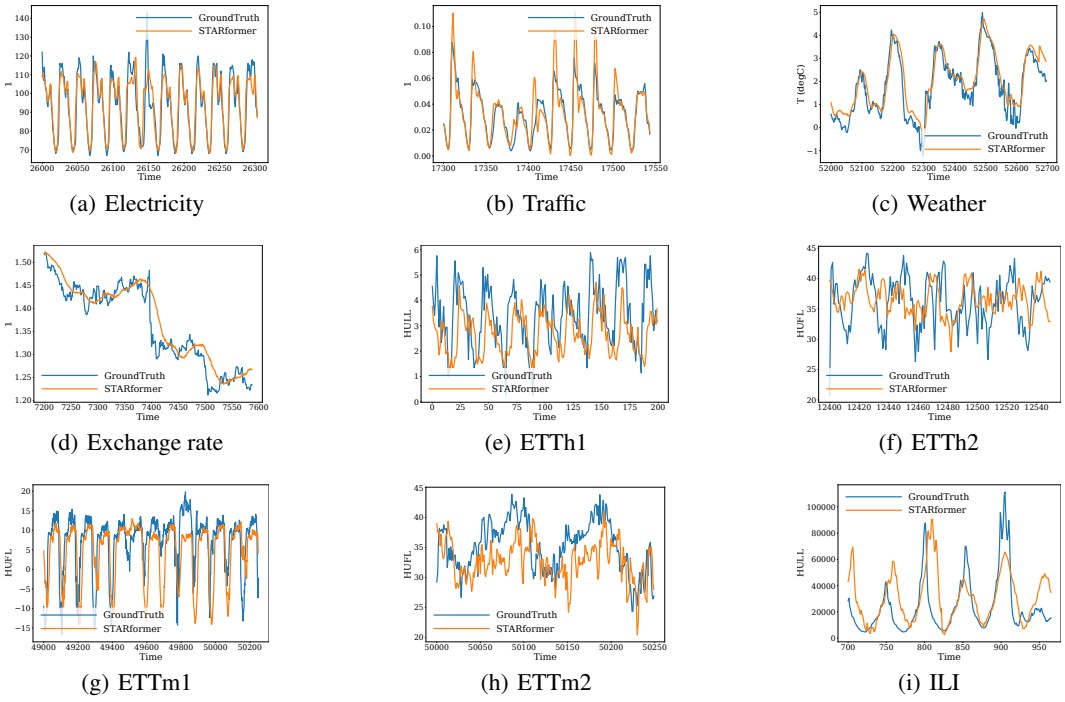

Figure 9: Visualization of STARformer forecasting results on 9 datasets.

Figure 10, the attention we propose mainly shows this increase/decrease and trend, although the intensity of attention is different for each columns. For example, looking at the exchange rate in Figure 10 (e), the second column from the top shows a linear trend. The attention (cf. Figure 10(f)) for this second column is close to 0, and the attention is strong only at a few points in between. On the other hand, in the case of the forth column from the top in Figure 10(e), changes in the actual data are frequently repeated, and the attention for the forth column shows a repeating pattern similar to the data.

### B.3 VISUALIZATION ON FORECASTING PERFORMANCE WITH VARYING INPUT SEQUENCES

In this subsection, we visualize our sensitivity analysis on the forecasting results according to the various input sequences. Table 2,3 and Table 8 show the results for the case where the input sequence $I$ and forecast horizon $H$ are the same. In Figure 6 and Figure 11 shows the experimental results on various input sequence $I \in \{48, 60, 72, 104\}$ for national illness and various input sequence $I \in \{48, 96, 192, 336, 720\}$. For the forecast horizon $H$, we fixed the shortest and the longest prediction length $H \in \{24, 60\}$ for national illness and $H \in \{96, 720\}$ for the other datasets. Overall, STARformer shows the lowest use results in all input sequences.

### B.4 VISUALIZATION OF TREND(OR SEASONAL PART) FORECASTING IN ALGORITHM 1

The core part of our model, STARformer, is structural attention which replaces self-attention. To calculate the structural attention of our model, we need to know $\hat{X}^{Simple}$ in Equation 4. As in Algorithm 1, we learn a single linear layer to obtain $\hat{X}^{Simple}$ to predict the future trend or future seasonal part. Figure 12 visualizes the trend and seasonal part predicted by the single linear layer with the actual data in blue at the top. In Figure 12, we visualize our single linear layer's forecasting results on 9 benchmarked datasets.

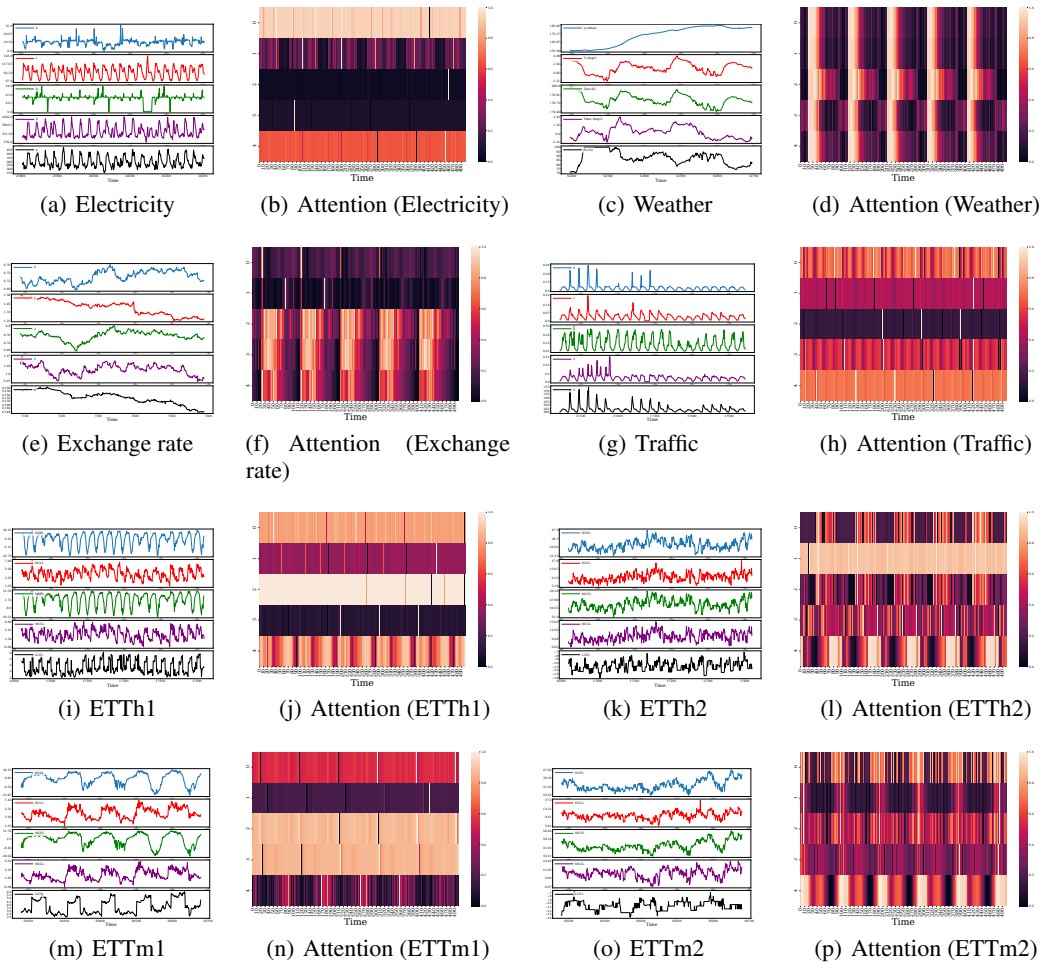

Figure 10: Visualization on attention heatmap on 8 benchmarked datasets. The brighter the red means strong attention, and the darker the red color means weak attention.

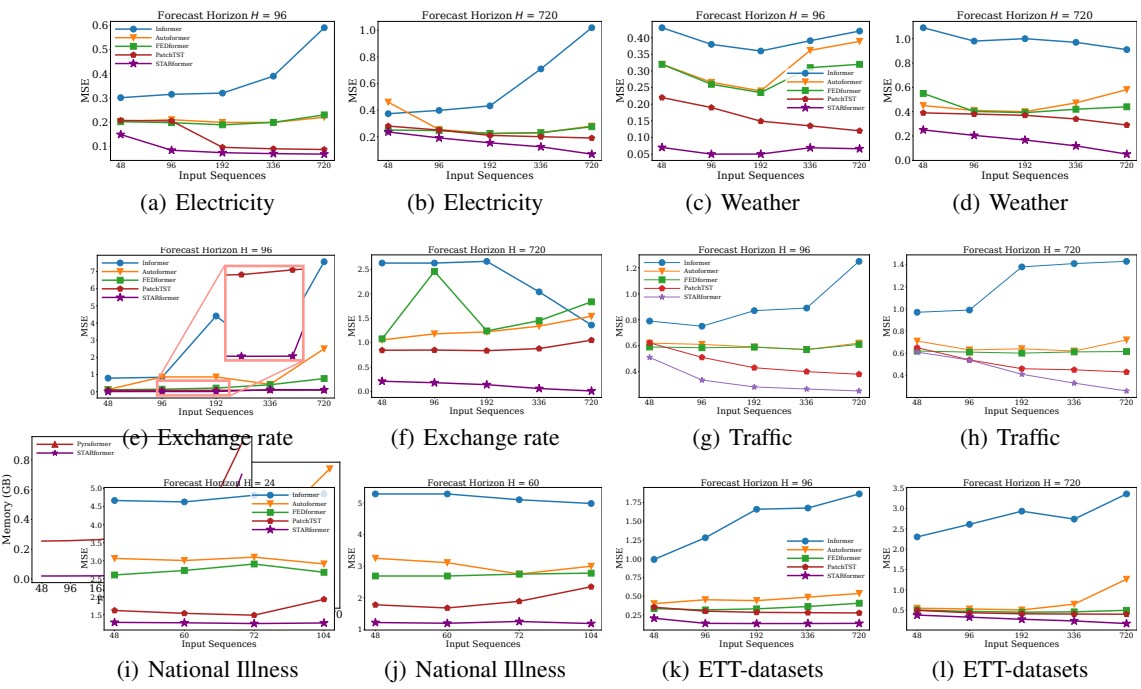

Figure 11: Forecasting performance (MSE) with varying input sequences on 2 large datasets: Electricity and Weather. Input sequence is $\{48, 96, 192, 336, 720\}$ and Forecast horizon is $\{96, 720\}$.

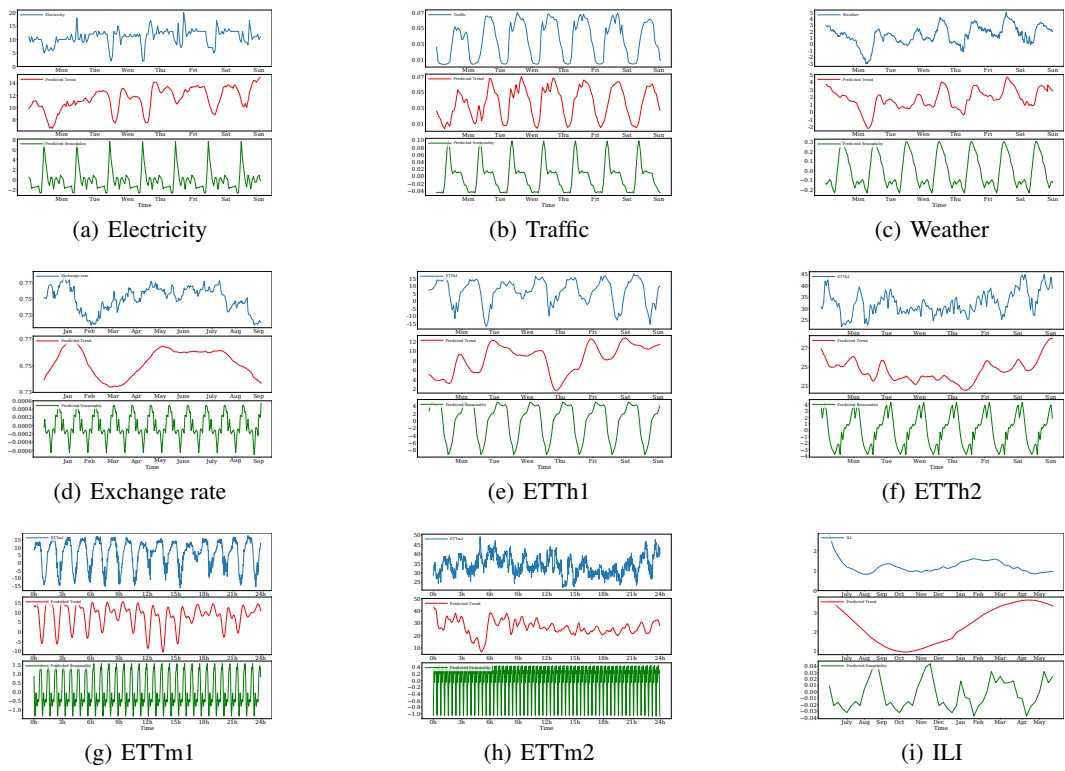

Figure 12: Visualization of single linear layer's prediction results on 9 datasets.

## C  ABLATION STUDY ON STRUCTURAL ATTENTION

Structural attention in STARformer is built from simple predicted time series. In this paper, simple time series are considered trends or seasonal parts. In other words, it is a method of predicting trends or seasonal parts and calculating attention from the predicted values. In Tables 2 and 3 and Table 8, we experimented with a simple time series fixed as a trend. Table 8 shows the results using attention generated from seasonal parts and compares them with state-of-the-art models of transformer-based and non-transformer-based models. Figure 12 visualizes real data (blue line), predicted trends (red line), and predicted seasonal parts (green line). Looking at the case of electricity, traffic, exchange rate, and ETTh1 in Figure 12, the seasonal characteristics seen in the data are clearly visible. Surprisingly, when seasonal parts are evident as shown in Table 7, STARformer (season) generally shows the second best performance. On the other hand, if seasonal characteristics are not representative of the data, such as Weather or National illness datasets, the performance is not good. The use of attention, which well represents the characteristics of the data, affects the experimental results of the model, and this supports the latest research trend in which time series prediction research, where attention was applied, has recently been replaced by attention methodologies based on time series analysis, such as Autoformer and ETSformer.

Table 7: Ablaton study on 9 benchmarked dataset

| Datasets | Electricity | | Traffic | | Weather | | Exchange | | ETTh1 | | ETTh2 | | ETTm1 | | ETTm2 | | | ILI | |
|---|---|---|---|---|---|---|---|---|---|---|---|---|---|---|---|---|---|---|---|
| horizons | MSE | MAE | MSE | MAE | MSE | MAE | MSE | MAE | MSE | MAE | MSE | MAE | MSE | MAE | MSE | MAE | horizons | MSE | MAE |
| **DLinear** 96 | 0.140 | 0.237 | 0.410 | 0.282 | 0.176 | 0.237 | 0.081 | 0.203 | 0.375 | 0.399 | 0.289 | 0.353 | 0.299 | 0.343 | 0.167 | 0.260 | 24 | 2.215 | 1.081 |
| 192 | 0.153 | 0.249 | 0.423 | 0.287 | 0.220 | 0.282 | 0.157 | 0.293 | 0.405 | 0.416 | 0.383 | 0.418 | 0.335 | 0.365 | 0.224 | 0.303 | 36 | 1.963 | 0.963 |
| 336 | 0.169 | 0.267 | 0.436 | 0.296 | 0.265 | 0.319 | 0.305 | 0.414 | 0.439 | 0.443 | 0.448 | 0.465 | 0.369 | 0.386 | 0.281 | 0.342 | 48 | 2.130 | 1.024 |
| 720 | 0.203 | 0.301 | 0.466 | 0.315 | 0.323 | 0.362 | 0.643 | 0.601 | 0.472 | 0.490 | 0.605 | 0.551 | 0.425 | 0.421 | 0.397 | 0.421 | 60 | 2.368 | 1.096 |
| **PatchTST** 96 | 0.129 | 0.222 | 0.360 | 0.249 | 0.149 | 0.198 | 0.093 | 0.218 | 0.370 | 0.400 | 0.274 | 0.336 | 0.290 | 0.342 | 0.165 | 0.255 | 24 | 1.319 | 0.754 |
| 192 | 0.147 | 0.240 | 0.379 | 0.256 | 0.194 | 0.241 | 0.208 | 0.332 | 0.413 | 0.429 | 0.339 | 0.379 | 0.332 | 0.369 | 0.220 | 0.292 | 36 | 1.007 | 0.870 |
| 336 | 0.163 | 0.259 | 0.392 | 0.264 | 0.245 | 0.282 | 0.359 | 0.440 | 0.422 | 0.440 | 0.329 | 0.384 | 0.366 | 0.392 | 0.274 | 0.329 | 48 | 1.553 | 0.815 |
| 720 | 0.197 | 0.290 | 0.432 | 0.286 | 0.314 | 0.334 | 1.194 | 0.815 | 0.447 | 0.468 | 0.379 | 0.422 | 0.416 | 0.420 | 0.362 | 0.385 | 60 | 1.016 | 0.788 |
| **STARformer season** 96 | 0.110 | 0.188 | 0.473 | 0.358 | 0.317 | 0.293 | 0.006 | 0.056 | 0.359 | 0.387 | 0.334 | 0.390 | 0.360 | 0.332 | 0.192 | 0.290 | 24 | 2.598 | 1.098 |
| 192 | 0.090 | 0.172 | 0.373 | 0.298 | 0.335 | 0.310 | 0.012 | 0.082 | 0.403 | 0.387 | 0.389 | 0.422 | 0.383 | 0.413 | 0.248 | 0.330 | 36 | 2.658 | 1.107 |
| 336 | 0.090 | 0.177 | 0.356 | 0.287 | 0.381 | 0.335 | 0.058 | 0.199 | 0.432 | 0.390 | 0.410 | 0.446 | 0.396 | 0.419 | 0.380 | 0.354 | 48 | 2.720 | 1.123 |
| 720 | 0.110 | 0.194 | 0.358 | 0.286 | 0.446 | 0.377 | 0.088 | 0.228 | 0.451 | 0.429 | 0.501 | 0.496 | 0.446 | 0.415 | 0.380 | 0.412 | 60 | 2.850 | 1.131 |
| **STARformer trend** 96 | 0.084 | 0.198 | 0.335 | 0.299 | 0.050 | 0.084 | 0.011 | 0.074 | 0.218 | 0.193 | 0.127 | 0.151 | 0.116 | 0.228 | 0.137 | 0.135 | 24 | 1.266 | 0.728 |
| 192 | 0.070 | 0.178 | 0.278 | 0.269 | 0.050 | 0.084 | 0.012 | 0.075 | 0.211 | 0.199 | 0.137 | 0.190 | 0.101 | 0.211 | 0.136 | 0.135 | 36 | 1.249 | 0.729 |
| 336 | 0.076 | 0.194 | 0.263 | 0.258 | 0.050 | 0.084 | 0.012 | 0.075 | 0.203 | 0.223 | 0.164 | 0.250 | 0.098 | 0.210 | 0.133 | 0.135 | 48 | 1.191 | 0.716 |
| 720 | 0.069 | 0.177 | 0.259 | 0.253 | 0.051 | 0.084 | 0.013 | 0.079 | 0.296 | 0.364 | 0.284 | 0.371 | 0.098 | 0.211 | 0.132 | 0.157 | 60 | 1.492 | 0.777 |

## D  COMPARED TO NON-TRANSFORMER MODELS

We also compare with non-transformer models (FiLM, NLinear, DLinear, N-Hits). Since STARformer uses time-series decomposition like N-Hits or DLinear, so we compare our method with non-transformer models as shown in Table 8. STARformer outperforms in all horizons on all 9 benchmarked datasets.

Table 8: Comparison of experimental results with non-transformer model on 9 benchmark datasets

| Datasets | horizons | Electricity MSE | MAE | Traffic MSE | MAE | Weather MSE | MAE | Exchange MSE | MAE | ETTh1 MSE | MAE | ETTh2 MSE | MAE | ETTm1 MSE | MAE | ETTm2 MSE | MAE | horizons | ILI MSE | MAE |
|---|---|---|---|---|---|---|---|---|---|---|---|---|---|---|---|---|---|---|---|---|
| DLinear 96 | | 0.140 | 0.237 | 0.410 | 0.282 | 0.176 | 0.237 | 0.081 | 0.203 | 0.375 | 0.399 | 0.289 | 0.353 | 0.299 | 0.343 | 0.167 | 0.260 | 24 | 2.215 | 1.081 |
| 192 | | 0.153 | 0.249 | 0.423 | 0.287 | 0.220 | 0.282 | 0.157 | 0.293 | 0.405 | 0.416 | 0.383 | 0.418 | 0.335 | 0.365 | 0.224 | 0.303 | 36 | 1.963 | 0.963 |
| 336 | | 0.169 | 0.267 | 0.436 | 0.296 | 0.265 | 0.319 | 0.305 | 0.414 | 0.439 | 0.443 | 0.448 | 0.465 | 0.369 | 0.386 | 0.281 | 0.342 | 48 | 2.130 | 1.024 |
| 720 | | 0.203 | 0.301 | 0.466 | 0.315 | 0.323 | 0.362 | 0.643 | 0.601 | 0.472 | 0.490 | 0.605 | 0.551 | 0.425 | 0.421 | 0.397 | 0.421 | 60 | 2.368 | 1.096 |
| NLinear 96 | | 0.141 | 0.237 | 0.410 | 0.279 | 0.182 | 0.232 | 0.089 | 0.208 | 0.374 | 0.394 | 0.277 | 0.338 | 0.306 | 0.348 | 0.167 | 0.255 | 24 | 1.683 | 0.858 |
| 192 | | 0.154 | 0.248 | 0.423 | 0.284 | 0.225 | 0.269 | 0.180 | 0.300 | 0.408 | 0.415 | 0.344 | 0.381 | 0.349 | 0.375 | 0.221 | 0.293 | 36 | 1.703 | 0.859 |
| 336 | | 0.171 | 0.265 | 0.435 | 0.290 | 0.271 | 0.301 | 0.331 | 0.415 | 0.429 | 0.427 | 0.357 | 0.400 | 0.375 | 0.388 | 0.274 | 0.327 | 48 | 1.719 | 0.884 |
| 720 | | 0.210 | 0.297 | 0.464 | 0.307 | 0.338 | 0.348 | 1.033 | 0.780 | 0.440 | 0.453 | 0.394 | 0.436 | 0.433 | 0.422 | 0.368 | 0.384 | 60 | 1.819 | 0.917 |
| FiLM 96 | | 0.154 | 0.267 | 0.416 | 0.294 | 0.199 | 0.262 | 0.079 | 0.204 | 0.371 | 0.394 | 0.284 | 0.348 | 0.302 | 0.345 | 0.165 | 0.256 | 24 | 1.970 | 0.875 |
| 192 | | 0.164 | 0.258 | 0.408 | 0.288 | 0.228 | 0.288 | 0.159 | 0.292 | 0.414 | 0.423 | 0.357 | 0.400 | 0.338 | 0.368 | 0.240 | 0.313 | 36 | 1.982 | 0.859 |
| 336 | | 0.188 | 0.283 | 0.425 | 0.298 | 0.267 | 0.323 | 0.270 | 0.398 | 0.442 | 0.445 | 0.377 | 0.417 | 0.373 | 0.388 | 0.286 | 0.345 | 48 | 1.868 | 0.896 |
| 720 | | 0.236 | 0.332 | 0.520 | 0.353 | 0.319 | 0.361 | 0.536 | 0.574 | 0.465 | 0.472 | 0.439 | 0.456 | 0.420 | 0.420 | 0.393 | 0.422 | 60 | 2.057 | 0.929 |
| N-Hits 96 | | 0.147 | 0.249 | 0.402 | 0.282 | 0.158 | 0.195 | 0.092 | 0.202 | 0.378 | 0.393 | 0.274 | 0.345 | 0.302 | 0.350 | 0.176 | 0.255 | 24 | 1.862 | 0.869 |
| 192 | | 0.167 | 0.269 | 0.420 | 0.297 | 0.211 | 0.247 | 0.208 | 0.322 | 0.427 | 0.436 | 0.353 | 0.401 | 0.347 | 0.383 | 0.245 | 0.305 | 36 | 2.071 | 0.934 |
| 336 | | 0.186 | 0.290 | 0.448 | 0.313 | 0.274 | 0.300 | 0.301 | 0.403 | 0.458 | 0.484 | 0.382 | 0.425 | 0.369 | 0.402 | 0.295 | 0.346 | 48 | 2.134 | 0.932 |
| 720 | | 0.243 | 0.340 | 0.539 | 0.353 | 0.401 | 0.413 | 0.798 | 0.596 | 0.472 | 0.561 | 0.625 | 0.557 | 0.431 | 0.441 | 0.413 | 0.413 | 60 | 2.137 | 0.968 |
| STARformer 96 | | **0.084** | **0.198** | **0.335** | **0.299** | **0.050** | **0.084** | **0.011** | **0.074** | **0.218** | **0.193** | **0.127** | **0.151** | **0.116** | **0.228** | **0.137** | **0.135** | 24 | **1.266** | **0.728** |
| 192 | | **0.070** | **0.178** | **0.278** | **0.269** | **0.050** | **0.084** | **0.012** | **0.075** | **0.211** | **0.199** | **0.137** | **0.190** | **0.101** | **0.211** | **0.136** | **0.135** | 36 | **1.604** | **0.739** |
| 336 | | **0.076** | **0.194** | **0.263** | **0.258** | **0.050** | **0.084** | **0.012** | **0.075** | **0.203** | **0.223** | **0.164** | **0.250** | **0.098** | **0.210** | **0.133** | **0.135** | 48 | **1.527** | **0.765** |
| 720 | | **0.069** | **0.177** | **0.259** | **0.253** | **0.051** | **0.084** | **0.013** | **0.079** | **0.296** | **0.364** | **0.284** | **0.371** | **0.098** | **0.211** | **0.132** | **0.157** | 60 | **1.492** | **0.777** |