# OpenReview forum: "STARformer: STructural Attention tRansformer for Long-term Time Series Forecasting"
_ICLR.cc/2024/Conference — ICLR 2024 Conference Withdrawn Submission_

### Official Review · Reviewer_DtH7 · 2023-10-31

**Soundness:** 1 poor
**Presentation:** 1 poor
**Contribution:** 2 fair
**Rating:** 1
**Confidence:** 4

**Summary:**

The authors introduce a new architecture for time series forecasting by replacing self-attention layers with linear models, aiming to discern the "structural attention" within time series data. Their method is two-staged: a coarse model makes an initial prediction, which a fine-tuning model then refines. This approach is both straightforward and computationally efficient. Experimental results highlight marked improvements over contemporary state-of-the-art models in long-term forecasting.

**Strengths:**

1. The presented architecture stands out for its simplicity and computational efficiency.
2. Experimental outcomes indicate notable superiority over existing state-of-the-art methodologies.

**Weaknesses:**

The overall presentation and structure of the paper currently fall short of the expected maturity for this submission. I recommend a major revision, taking into account the following key points:

1. **Clarity in Methodology**: The underlying advantages and rationale of the proposed method, especially in contrast to self-attention, need to be better delineated. It's imperative to provide a lucid exposition on the essence and necessity of each component in the proposed architecture.

2. **Literature Review**: The discussion on related works appears insufficient. It's crucial to acknowledge and distinguish from significant contributions like DLinear[1], which emphasizes the utility of linear models in time series forecasting. Similarly, referencing works like TSMixer[2], which have explored replacing self-attention with linear layers, can offer richer context.

3. **Definitions and Variables**: Ensure that all variables are introduced systematically and are well-defined. Premature introduction (e.g., $t, H, I$) or lack of definitions (e.g., $ B_{t-I:t}, X^{Simple}_{t-I:t}, \theta_f, Linear(\cdot), L_Simple $ ) hampers the readability and understanding of the paper.

4. **Equations and Formalism**: The paper needs more rigor in its mathematical formalism. Ambiguities, like the mislabeling in Eq (4), should be rectified. Moreover, providing explicit rules or guidelines for matrix operations can help eliminate any confusion.

5. **Complexity Analysis**: The complexity analysis appears to be flawed. Linear models also exhibit a quadratic complexity relative to the lengths. Current assessments, such as the interpretation of time growth in Fig 8, seem misleading, as external factors like GPU parallelism could influence the results.

6. **Experimental Results**: The results derived from the Weather dataset looks doubtful. Consistent results across different forecast horizons make them suspect. Providing access to the code would be beneficial for verification purposes.

[1] Zeng, Ailing, et al. "Are transformers effective for time series forecasting?." Proceedings of the AAAI conference on artificial intelligence. Vol. 37. No. 9. 2023.

[2] Chen, Si-An, et al. "TSMixer: An All-MLP Architecture for Time Series Forecasting." Transactions on Machine Learning Research. 2023

**Questions:**

1. In the case of a $d$-variate time series, does the proposed model account for cross-variate correlations or treat each variate as isolated?
2. Are there any non-linear transformations embedded within the proposed method? Merging several linear models would essentially replicate the effect of a singular linear layer.
3. Could the authors elucidate the concepts of "encode" and "decode" within their framework? Is it necessary to be trained in two-stage, or can it be trained end-to-end as a single network?

---

### Official Review · Reviewer_9etD · 2023-11-01

**Soundness:** 3 good
**Presentation:** 2 fair
**Contribution:** 3 good
**Rating:** 6
**Confidence:** 3

**Summary:**

This paper proposes structural attention Transformer for time-series forecasting. It decomposes time series into trend and seasonality components and learns the trend and seasonality patterns using a pre-trained linear layer. The learned attention guides the model to correctly focus on the input sequence. The proposed method is shown to outperform existing baselines by a large margin.

**Strengths:**

1. This paper presents a new structural attention by decomposing time series into trend and seasonality, and learn these simpler patterns with pre-trained linear layer.
2. Experiments on benchmark forecasting datasets show significant improvement compared with the baselines.

**Weaknesses:**

1. The paper needs to clarify the formal definition of X^{Simple} and discuss methods used to assess the importance of trend and seasonality within the overall data representation. The ablation study shows the effect of separately having trend and seasonality. Will a combined approach of learning both components lead to better performance?
2. It remains unclear why errors on weather and exchange rate datasets remain constant across different forecasting horizons. Questions also arise regarding the training methodology of the single-layer linear function f - whether it's pre-trained independently or trained end to end with other parts of the encoder and decoder? More details about training the structural attention (such as data split, decomposition) should be discussed to ensure there is no data leakage problem when training the linear model.
3. Why the proposed model has more smooth loss landscape? How does it compare to the landscape of other linear models like DLinear?
4. S_{t:t+H} is essentially a linear mapping respect to trend or seasonality. Mathematically what are the differences between learning structural attention and learning a direct linear mapping from decomposed trend and seasonality components of the original data?

**Questions:**

See Weaknesses above.

---

### Official Review · Reviewer_XxxW · 2023-11-03

**Soundness:** 2 fair
**Presentation:** 1 poor
**Contribution:** 1 poor
**Rating:** 1
**Confidence:** 4

**Summary:**

This paper proposes a new

**Strengths:**

Sufficient empirical experiments are performed and reported with analysis on the results.

**Weaknesses:**

1. The paper needs significant efforts to improve clarity and reasoning. Most of the methodology is difficult to understand due to lack of clarification of used notations / terminologies and details in steps. Therefore, I am afraid I don't understand the main idea the authors want to introduce. See details in questions.
2. As far as I can see (after checking the provided source code), the proposed method is trivial compared to previous models. Based on the operations listed in sec 3.2 and implementations,  it's essentially a linear model based on conventional time series decomposition. The motivation behind the designs are quite weak with neither strong intuition nor persuasive eidences, which makes the empirical results not convincing.

**Questions:**

1. What is *structural attention*? Is this a well-known concept as described in this paper? From my perspective, it is the linear *weight matrix* of a linear model as eq (1). From eq (2), I can see $S_{t:t+H}$ is the weight of a new linear model by replacing the independent variable and intercept in $f$. What is the point of such transformation? In addition, since self-attention is substituted by such a linear model, it is not appropriate to name the method as "Transformer".
2. The algorithm 1 is really confusing. Is $X_{simple}$ from trend or seasonality, or both? Which time series decomposition method is used? And I don't think such a trivial linear model training procedure needs to be highlighted.
3. Are the linear layers in eq (5) (7) (8) different in terms of parameters? Are $\alpha, \beta, \gamma$ in eq(9) scalars or vectors? How do you initialize and regularize them?

---

### Official Review · Reviewer_qcgb · 2023-11-03

**Soundness:** 2 fair
**Presentation:** 1 poor
**Contribution:** 1 poor
**Rating:** 3
**Confidence:** 3

**Summary:**

The paper proposes a long-term time series model that is a weighted combination of three parts: (1) an encoder block that uses what the authors call "structural attention" to map the past into the horizon (ii) a linear map of trend component of the past into the horizon and (iii) a linear map from (past - trend) into the horizon. It is not clear to me after reading the paper and the code what exactly the encoder block does -- see questions below. The paper displays surprisingly good performance compared to prior SOTA models like PatchTST even though it misses comparison in terms of computational complexity w.r.t more efficient MLP based models.

**Strengths:**

1. The numerical results presented in Table 2 and 3 are quite impressive though a bit unbelievable to me as I do not fully understand how the representation power of the model is different from the DLinear model.

2. Though not surprising the computational complexity of the model is better than that of transformers as shown in Figure 8. This is due to the fact that the model uses only linear components to the best of my understanding.

**Weaknesses:**

1. The description of the encoder block which seems to be the main differentiator of the model is extremely confusing. I could not understand the main functional equations even after reading the Encoder block code provided in a link provided in the paper. I am assuming that the operations in Equation 1 are element wise as otherwise the dimensionality is incorrect. Even assuming that,  it is not clear what the definition of $B_{t-I:t}$ is. Is it learnt or just extracted from the data? Why is the matrix $S_{t:t+H}$ even called attention. It has no similarity to attention mechanism and just seems to be a element wise division.

The equations in section 3.2 where $H \neq I$ is even more confusing. The authors never define a functional map from $X_{t-I:t}$ to $\hat{X}_{t:t+H}^{Simple}$. The latter is never defined.

2. The mapping  of Trend and Seasonality to the future predictions in the decoder part seems to be exactly the same as that of the DLinear model. Is that correct?

3. Are the weights $\alpha, \beta$ and $\gamma$ learnt during training? How are they set if not learnt?

4. Given that the encoder part is not understandable and the two other components are exactly the same as that of DLinear I find the performance a bit unbelievable. The code is also not very easy to read as I don't know what "IBM" means as an input to the forward function. In Fig 1 can you also plot the predictions from DLinear.

5. Why is the structural attention called "attention" when it seems to be a linear map and there is not attention mechanism?

6. The paper talks about efficiency but only compares time complexity to other transformer models. There are more efficient MLP based models like TiDE[1], NHiTS that achieve similar performance but are much faster. It would be good to compare to and cite these works, in terms of efficiency and statistical performance.

[1] Das A, Kong W, Leach A, Mathur S, Sen R, Yu R. Long-term Forecasting with TiDE: Time-series Dense Encoder.

**Questions:**

I have asked the main questions in the previous section.